# Comparative Transcriptome Analysis Identified Potential Genes and Transcription Factors for Flower Coloration in Kenaf (*Hibiscus cannabinus* L.)

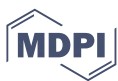

Jae Il Lyu [1,2,†], Jaihyunk Ryu [3,†], Dong-Gun Kim [3], Jung Min Kim [3], Joon-Woo Ahn [3], Soon-Jae Kwon [3], Sang Hoon Kim [3] and Si-Yong Kang [1,*]

1   Department of Horticulture, College of Industrial Sciences, Kongju National University, Yesan 32439, Republic of Korea
2   Department of Agricultural Biotechnology, National Institute of Agricultural Sciences, RDA, Jeonju 54874, Republic of Korea
3   Advanced Radiation Technology Institute, Korea Atomic Energy Research Institute, Jeongeup 56212, Republic of Korea
*   Correspondence: sykang@kongju.ac.kr
†   These authors contributed equally to this work.

**Abstract:** The biochemical compounds in kenaf leaves and flowers mainly consist of flavonoids, including flavonoid glycosides and floral anthocyanins. In the present study, we performed comparative transcriptome analysis using RNA-sequencing and identified putative genes involved in flower coloration in different flower developmental stages of three kenaf mutants including Baekma (white flower), Jangdae (ivory flower), and Bora (purple flower). A total of 36.1 Gb reads were generated for two kenaf accessions and 38,601 representative transcripts with an average length of 1350 bp were yielded, of which 33,057 (85.64%) were annotated against two databases. Expression profiling of the transcripts identified 1044 and 472 differentially expressed genes (DEGs) among three mutants in the young bud and full bloom stages, respectively. KEGG enrichment analysis of these DEGs revealed that the representative pathway was "biosynthesis of secondary metabolites", including phenylpropanoid biosynthesis and flavonoid biosynthesis. Consequently, we investigated genes related to the phenylpropanoid pathway, which included 45 DEGs from a total of 1358. Our results provide useful information for understanding gene functions of flower coloration in kenaf, which will be useful in further studies.

**Keywords:** kenaf; RNA-seq; comparative transcriptome analysis; flower



## 1. Introduction

Kenaf (*Hibiscus cannabinus* L.), an annual fiber plant belonging to the Malvaceae family, is an important medicinal crop as it contains high amounts of phytochemical compounds including polyphenols, flavonoids, and anthocyanins. Accordingly, kenaf has been used in African and Indian traditional medicine as an aphrodisiac, purgative, and digestive aid [1]. Furthermore, some studies reported that kenaf has other medicinal properties including antioxidant, hepatoprotective, and anticancer activities [2–4].

Flavonoids and anthocyanins are not only phytochemical compounds but also natural pigments in plants. The biosynthesis of flavonoids is included in the phenylpropanoid pathway, which comprises several biosynthetic branches such as lignin, stilbenes, and anthocyanins synthesis [5]. The enzymes involved in this pathway include phenylalanine ammonia lyase (PAL), cinnamate 4-hydroxylase (C4H), 4-coumarate-CoA ligase (4CL), chalcone synthase (CHS), chalcone isomerase (CHI), flavone 3-hydroxylase (F3H), dihydroflavonol 4-reductase (DFR), and flavone synthase (FLS). Generally, the genes encoding these enzymes, which are commonly called the "structural genes" in phenylpropanoid biosynthetic pathway, are regulated by various transcription factors such as WRKY, bHLH,

and R2R3-MYBs [6]. MYB transcription factors are one of the largest plant transcription factor families, among which, R2R3-MYBs regulate the synthesis of phenylpropanoid-derived compounds [7]. In Arabidopsis thaliana, PAP1/MYB75 and PAP2/MYB90, a member of subgroup 6 in R2R3-MYBs, are regulators of genes required for the production of *PAL*, *CHS*, *DFR*, and glutathione-s-transferase (*GST*) [8]. *MYB11*, *MYB12*, and *MYB111* are members of subgroup 7 in R2R3-MYB control flavonol biosynthetic genes including *CHS*, *CHI*, *F3H*, and *FLS1* [9].

In kenaf plant, the genes of phenylpropanoid biosynthetic enzymes including *HcPAL*, *Hc4CL*, *HcC4H*, and *HcCHS* have been isolated in previous studies [10–12] and their expression patterns were evaluated in different organs [13]. Additionally, the recent progress in sequencing methods, including NGS technology, has facilitated various transcriptome analysis studies in kenaf. For example, in a comparative transcriptome analysis for identifying genes related to the biosynthesis of anthocyanins and kaempferitrin in kenaf leaves, 29 DEGs were assigned to 8 structural genes including *4CL*, *CHS*, *CHI*, *F3H*, *DFR*, *ANS*, *FLS*, and *3GT* [14]. Using the PAC-Bio sequencing method, the novel candidate genes were found to be related to the lignin and cellulose biosynthetic pathways in kenaf leaves [15]. Moreover, salinity-responsive genes [16], protein synthesis genes [17], and cadmium-responsive genes [18,19] were explored by transcriptome analysis through RNA-seq methods. However, these studies mainly investigated the roots, stems, and leaves of kenaf, whereas studies on flowers were limited.

Previously, we successfully investigated the transcriptome analysis of leaf coloration in kenaf, which exhibited dramatic changes in eight flavonoid structural genes. The present study aimed to examine the content of phenolic compounds and perform transcriptome analysis on three different kenaf flowers including the Baekma (BM), Jangdae (JD), and Bora (BR). Furthermore, the potential genes involved in kenaf flower coloration were explored using the results of the comparative transcriptome analysis. These findings will contribute to understanding the molecular mechanism of flower pigmentation in kenaf.

## 2. Materials and Methods

### 2.1. Plant Materials and Biochemical Analysis

Three kenaf mutants, Baekma (white petal), Jangdae (ivory petal), and Bora (purple petal), were used in the present study. The Baekma is a natural mutant variety originating from C-14. Jangdae and Bora are gamma-ray-induced mutant varieties originating from Jinju and Hongma300, respectively. The kenaf mutants were grown in a greenhouse at the Korea Atomic Energy Research Institute (KAERI, Jeongeup, Republic of Korea) under natural conditions.

The phenolic compound contents of full-bloomed flowers were analyzed and quantified using Ultra-Performance Liquid chromatography (UPLC) following the method of a previous study [14].

### 2.2. RNA Extraction and cDNA Library Construction

Total RNA was extracted from flower samples of the two different developmental stages using APureTM Plant SFGR RNA kit (AP BIO, Namyangju, Republic of Korea), following the manufacturer's protocol. RNA quantity and integrity were evaluated on a Nanodrop ND-1000 (Thermo Fisher Scientific, Waltham, MA, USA) and 2100 Bioanalyzer (Agilent, Santa Clara, CA, USA). RNA-seq paired-end libraries were prepared using the Illumina TruSeq RNA Sample Preparation Kit v2 (Illumina, San Diego, CA, USA). The library was quantified with the KAPA library quantification kit (Kapa biosystems, Wilmington, MA, USA) following the manufacturer's instructions. Each library was loaded into the Illumina HiseqX platform, and high-throughput sequencing was performed to ensure that each sample meets the desired average sequencing depth. RNA-seq raw reads data are available in NCBI Sequence Read Archive (SRA) under the following accession numbers: SRR23095825-SRR23095836.

### 2.3. De Novo Assembly and Annotation

The duplicated read produced by PCR was filtered through in-house scripts. Sequencing data with qualities (Q) $\geq$ 20 were extracted by SolexaQA [20]. Clean reads were de novo assembled into transcripts using Trinity with default parameters [21]. Using the clean reads merged from each sample, transcripts were validated by direct comparison with gene sequences from the InterProscan and NCBI NR viridiplantae database by BLASTX algorithms with a significant threshold of e-value $\leq 1 \times 10^{-10}$. The proteins with the highest sequence similarities were retrieved for analysis.

### 2.4. Differentially Expressed Genes (DEG) and Functional Enrichment Analysis

The number of mapped clean reads for each transcript was calculated and normalized using the DESeq package in R [22]. The fold changes and the number of reads mapped on unigene were used to identify differentially expressed genes in each sample. FDR (false discovery rate) was calculated via DESeq and used to identify the threshold of the *p*-value in multiple tests and analyses. Correlation analysis and hierarchical clustering were performed using AMAP library in R [23]. GO (Gene Ontology) and KEGG (Kyoto Encyclopedia of Genes and Genomes) enrichment analyses were carried out based on the sequence similarity (e-value cut off $\leq 1 \times 10^{-10}$) of each protein in the databases.

### 2.5. Quantitative Real-Time PCR Analysis

First-strand cDNA synthesis was carried out using 0.5 μg of total RNA as a template, along with SuperScript III First-Strand Synthesis SuperMix (Invitrogen, USA). DEG-specific primers (Table S1) were designed for the selected transcript sequence using the Primer3 software (http://www.bioinformatics.nl/cgi-bin/primer3plus/primer3plus.cgi: accessed on 1 February 2023). qRT-PCR was performed using iTaq Universal SYBR Green SuperMix (Bio-Rad, Hercules, CA, USA) and the Bio-Rad CFX96 real-time PCR detection system. Gene expression levels were normalized against the expression of the actin (*HcACT7*) gene [24] and calculated based on the $2^{-\Delta\Delta Ct}$ comparative threshold method [25].

## 3. Results

### 3.1. Composition of the Phenolic Compounds in Three Kenaf Flowers

The three newly developed kenaf mutants, Baekma (BM), Jangdae (JD), and Bora (BR), showed white, ivory, and purple petals, respectively (Figure 1a). Generally, the flower color is affected by the content of phenolic compounds, including flavonoids and anthocyanins. To understand the biochemical status of kenaf flower according to the color variation, phenolic compounds, including total phenolic content (TPC), total anthocyanin content (TAC), hibiscus acid, and three different anthocyanins, were measured using the UPLC system. As shown in Table 1, phenolic compounds varied among the three kenaf flowers. TPC (132.98 $\pm$ 12.56 mg/g FW) and TAC (48.85 $\pm$ 6.20 mg/g FW) levels were significantly higher in BR compared with JD (TPC; 96.81 $\pm$ 2.45 mg/g FW, TAC; 19.90 $\pm$ 1.36 mg/g FW) and BM (TPC; 63.14 $\pm$ 2.95 mg/g FW, TAC; 12.60 $\pm$ 0.32 mg/g FW). Similarly, the hibiscus acid levels were also higher in BR (12.29 $\pm$ 0.06 mg/g FW) than JD (6.13 $\pm$ 0.42 mg/g FW) and BR (8.99 $\pm$ 0.78 mg/g FW). Three different anthocyanins were detected in BR flower: delphinidin-3-sambubioside (D3S; 16.22 $\pm$ 1.48 mg/g FW), delphinidin-3-galactoside (D3G; 14.65 $\pm$ 2.34 mg/g FW), and quercetin-3-glucoside (Q3G; 50.90 $\pm$ 0.54 mg/g FW). The D3G and D3S contents were 40–50% lower in JD and BM compared to BR flowers. These results indicate that purple BR flowers exhibited higher phenolic compound contents compared with white flowers, and these compounds contributed to coloration of kenaf flowers.

### 3.2. Transcriptome Analysis

To evaluate the molecular basis of flower color variation in three kenaf mutants, transcriptome analysis among three flower types and two different developmental stages (young bud and full bloom) was performed with two biological replicates for each sample. A total of 39.2 Gb of raw reads were generated in twelve libraries. After pre-processing

to remove low-quality reads (upper than Q $\geq$ 20), 36.2 Gb clean reads were de novo assembled into transcripts (Table S2). A total of 81,982 transcripts and 38,601 representative transcripts were discovered in kenaf mutants with an N50 score of 1882 bp and 1643 bp, respectively. Among the 38,601 representative transcripts, 33,057 (85.64%) had BLASTX hit in the InterProscan and NCBI NR viridiplantae database (Table 2).

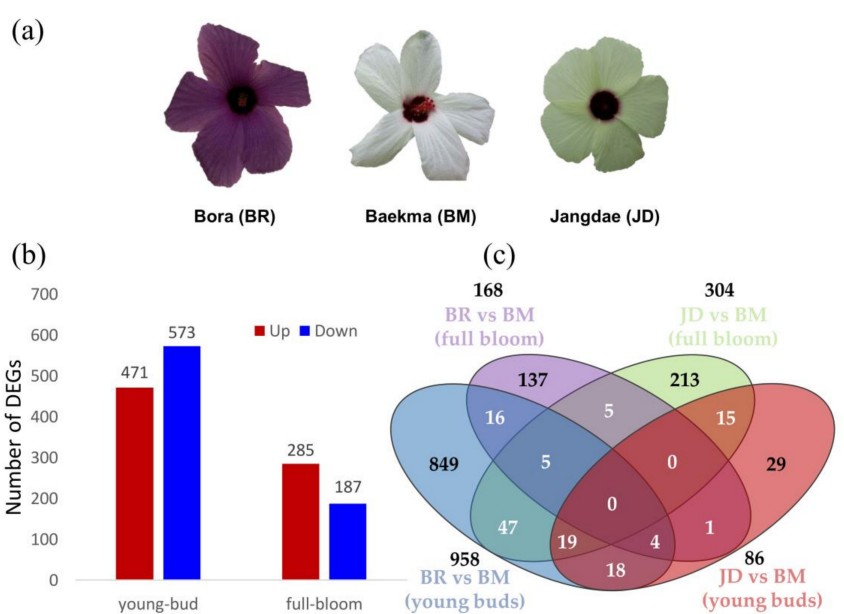

**Figure 1.** Variation of kenaf flower and identification of differentially expressed genes (DEGs) among three kenaf mutants: Bora (BR), Baekma (BM), Jangdae (JD). (**a**) Flower color variation in petals, (**b**) Up- and down-regulated DEGs compared with BM at young-bud and full-bloom stages. (**c**) Venn diagram of common and unique DEGs in the four indicated comparison groups.

**Table 1.** Contents of phenolic compounds in three different kenaf flowers.

| No. | Retention Time (min) | Identification | Molecular Formula | Negative ion Mass | Ref. |
|---|---|---|---|---|---|
| 1 | 3.9 | Hibiscus acid | $C_6H_5O_7$ | 207 | [1,2] |
| 2 | 7.8 | Delphinidin 3-sambubioside (Hibiscin) | $C_{26}H_{27}O_{16}$ | 595 | [1,2] |
| 3 | 8.2 | Delphinidin 3-galactoside | $C_{21}H_{19}O_{12}$ | 463 | [1,2] |
| 4 | 12.9 | Quercetin 3-glucoside | $C_{21}H_{19}O_{12}$ | 463 | [2] |

| | TPC [1] | TAC [2] | Hibiscus acid | D3S [3] | D3G [4] | Q3G [5] |
|---|---|---|---|---|---|---|
| Baekma | 63.14 ± 2.95 | 12.60 ± 0.32 | 8.99 ± 0.78 | 8.32 ± 1.58 | 0.00 | 1.29 ± 0.13 |
| Jangdae | 96.81 ± 2.45 ** | 19.90 ± 1.36 ** | 6.13 ± 0.42 ** | 9.75 ± 1.99 | 0.00 | 54.10 ± 0.51 ** |
| Bora | 132.98 ± 12.56 ** | 48.85 ± 6.20 ** | 12.29 ± 0.06 ** | 16.22 ± 1.48 ** | 14.65 ± 2.34 | 50.90 ± 0.54 ** |

[1] TPC: Total phenolic content, [2] TAC: Total anthocyanin content, [3] D3S: Delphinidin 3-sambubioside (Hibiscin), [4] D3G: Delphinidin 3-galactoside, [5] Q3G: Quercetin 3-glucoside. ** Significant at $p < 0.01$.

**Table 2.** Summary of assembled gene set and functional annotation.

| Data | Num. of Transcripts | Length (bp) of Transcripts | | | | | Functional Annotation | | |
|---|---|---|---|---|---|---|---|---|---|
| | | Sum. of bp | Min | Max | Average | N50 | NR Viridiplantae | Interproscan | Total Annotation |
| Total transcripts | 81,982 | 127,443,212 | 500 | 15,841 | 1554 | 1882 | 73,666 (89.86%) | 58,963 (71.92%) | 73,808 (90.03%) |
| Representative transcripts | 38,601 | 52,149,684 | 500 | 15,841 | 1350 | 1643 | 32,960 (85.39%) | 25,638 (66.42%) | 33,057 (85.64%) |

### 3.3. Comparison of DEGs among the Three Different Kenaf Flowers

To identify the differentially expressed genes (DEGs) among three mutant flowers, comparative analysis identified 1044 (471 up-regulated and 573 down-regulated) and 472 (285 up-regulated and 187 down-regulated) differentially expressed genes (DEGs) among three mutants at young buds and full bloom stages, respectively ($p < 0.05$ and log2-FC > 1) (Figure 1b). More detailed, as a young-bud stage, 958 DEGs were uncovered in BR vs. BM, while only 86 DEGs were identified in JD vs. BM. On the other hand, as a full-bloom stage, JD vs. BM (304 DEGs) had slightly more identified DEGs than BR vs. BM (168 DEGs). A Venn diagram showed the shared and specific number of DEGs among three mutant flowers within two developmental stages (Figure 1c). Additionally, collected DEGs were performed bioinformatic analysis through heatmap along with hierarchical clustering and functional enrichment analysis for better understanding DEGs' function. As a heatmap and clustering results (Figure 2a), these showed distinct expression patterns among three kenaf flowers on the different developmental stages and 1358 DEGs (removed duplicates from 1516 total DEGs) divided into six groups. Among the six groups, the C1 and C4 groups were found to be major DEG groups containing 837 DEGs. Based on the clustering analysis, the functional enrichment analysis was separately performed for each group by Gene Ontology (GO) and KEGG databases. From the GO enrichment analyses, the C1 group including 437 DEGs was enriched in 'carbohydrate metabolic process' and 'ion transport' (BP terms); 'catalytic activity' and hydrolase activity' (MF terms); and 'membrane and cell periphery' (CC terms). The C4 group, including 400 DEGs, was enriched in 'the response to chemical and response to organic substance' (BP terms); 'glucosyltransferase activity,' 'hydrolase activity', and 'acting on glycosyl bond's (MF terms); 'extracellular region; and 'cell wall' (CC terms). More detailed information for GO term enrichment analysis is described in Table S3.

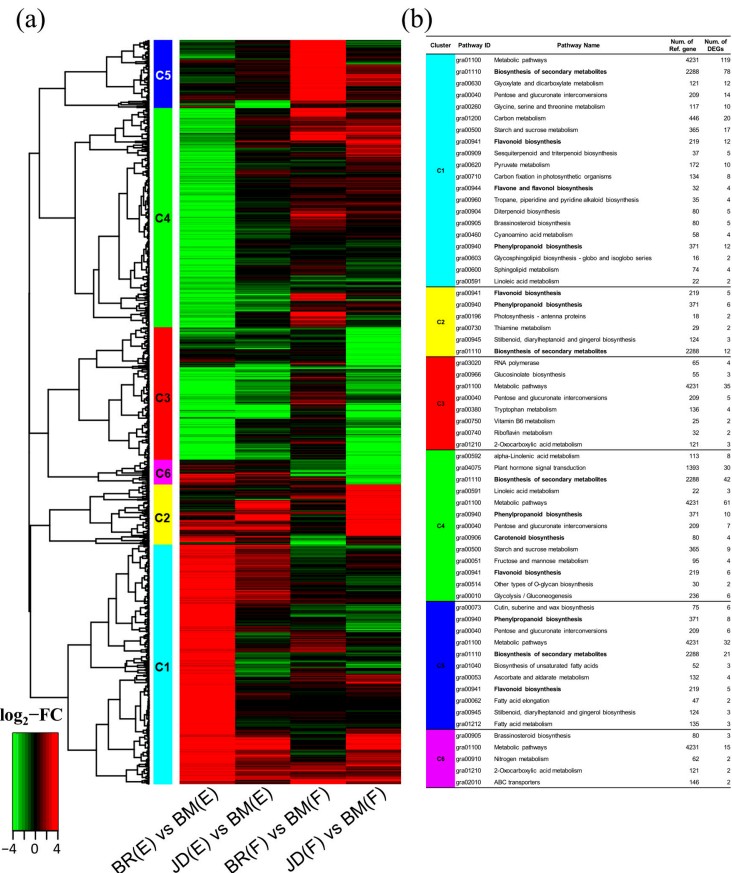

**Figure 2.** Differential expression analysis among comparison groups. (**a**) Hierarchical clustering, (**b**) KEGG enrichment analysis.

The KEGG enrichment analysis was also performed for each clustering group. As a result, most DEGs were associated with the biosynthesis of secondary metabolites, phenylpropanoid biosynthesis, and flavonoid biosynthesis (Figure 2b). For example, DEGs of the C1, C2, C4, and C5 groups are consistently associated with the phenylpropanoid pathway. These results suggest the involvement of phenylpropanoid pathway genes in kenaf flower color variation, as well as in other plants.

### 3.4. Identification of Putative Genes Involved in the Phenylpropanoid Pathway

As shown by the KEGG enrichment analysis of DEGs, the phenylpropanoid pathway was strongly enriched. Therefore, analysis of in phenylpropanoid pathway gene is required for a better understanding of flower color variation. Among the 1358 DEGs, a total of 45 DEGs were mapped to the entire pathway (Figure 3). These 45 DEGs were assigned to 24 structure genes, including 5 genes of flavonoid biosynthetic pathway: chalcone synthase (*CHS*), flavonoid 3′-monooxygenase (*F3′H*), flavonol synthase (*FLS1*, *FLS2*), dihydroflavonol 4-reductase (*DFR*); 8 genes of anthocyanin biosynthetic pathway: 5-O-glucoside-6″-O-malonyltransferase (*5MAT*), UDP-glucose:flavonoid 3-O-glucosyltransferase (*UF3GT*), UDP-glucosyltransferase (*UGT73B2*, *UGT75C1*, *UGT73C6*), transparent testa (*TT12*) glutathione S-transferase (*GST26*), ribonuclease (*RNS1*); and 10 genes of lignin biosynthetic pathway: coumarate 3-hydroxylase (*C3H*), 4-hydroxycinnamoyl CoA:shikimate/quinate hydroxycinnamoyltransferase (*HCT*), cinnamoyl CoA reductase (*CCR1*), cinnamyl alcohol dehydrogenase (*CAD8*, *CAD9*), beta-glucosidase (*BGLU10*, *BGLU45*, *BGLU46*), irregular xylem (*IRX*), UDP-glucosyltransferase (*UGT72E1*, *UGT72E2*). The gene encoding *CHS* (TRINITY_DN2412_c0_g1_i1, TRINITY_DN2412_c0_g2_i2) and *F3′H* (TRINITY_DN12895_c0_g4_i1) were up-regulated in BR mutant at the full-bloom and young-bud stages, respectively. In addition, flavonols and anthocyanidins synthase genes such as *FLS1* (TRINITY_DN14944_c0_g1_i1) were consistently up-regulated in BR and JD mutants at both developmental stages compared with the BM mutant. A total of six DEGs of anthocyanins transferase genes including *5MAT* (TRINITY_DN6647_c0_g1_i1), *UFGT* (TRINITY_DN21246_c0_g2_i1), *GST26* (TRINITY_DN5008_c0_g1_i8), *UGT75B2* (TRINITY_DN12056_c0_g2_i1), and *UGT75C1* (TRINITY_DN10696_c0_g1_i1, TRINITY_DN6166_c0_g3_i1) were involved in the acylation, glycosylation, and methylation of anthocyanins. Their expression levels were strongly increased in the BR mutant in the young-bud stage.

Furthermore, the most mapped DEGs in the phenylpropanoid pathway detected by comparing BR and BM mutants included 27 DEGs (young-bud) and 8 DEGs (full-bloom), respectively. On the other hand, seven DEGs were differentially expressed only in JD, compared to BM and were consistently distributed in the lignin biosynthetic pathway as TFs. One gene variant (TRINITY_DN269_c0_g3_i1) encoding DFR was up-regulated in JD vs. BM at both developmental stages (Figure 3). Therefore, these findings suggest that the identified DEGs involved in the flavonoid and anthocyanin biosynthetic pathway may play significant roles in the development of purple coloration of the petals of kenaf flowers.

Additionally, the expression pattern of TFs involved in phenylpropanoid pathway were identified. A total of 12 DEGs related to TFs were detected from 45 DEGs and were assigned to 11 TF genes involved in phenylpropanoid pathway; *WRKY75* (TRINITY_DN14705_c0_g1_i2, TRINITY_DN28549_c0_g1_i1), R2R3/or R3 *MYB* family genes (TRINITY_DN13448_c0_g2_i2, TRINITY_DN2082_c0_g2_i1, TRINITY_DN4288_c0_g1_i4, TRINITY_DN449_c0_g1_i3, TRINITY_DN449_c1_g1_i4, TRINITY_DN474_c0_g1_i2, TRINITY_DN474_c0_g2_i2, TRINITY_DN5866_c0_g2_i2), and *PIF3* (TRINITY_DN10758_c0_g1_i1, TRINITY_DN6710_c0_g2_i1) (e-value $\leq 1 \times 10^{-10}$, identity $\geq 50$). Among these putative genes of TFs, seven transcripts were strongly down-regulated in BR vs. BM at the young-bud stage, whereas two transcripts were significantly up-regulated in BR, compared to BM, at the full-bloom stage (Figure 3). As with the DEG expression pattern in structure genes, the TF genes were mainly involved in petal coloration.

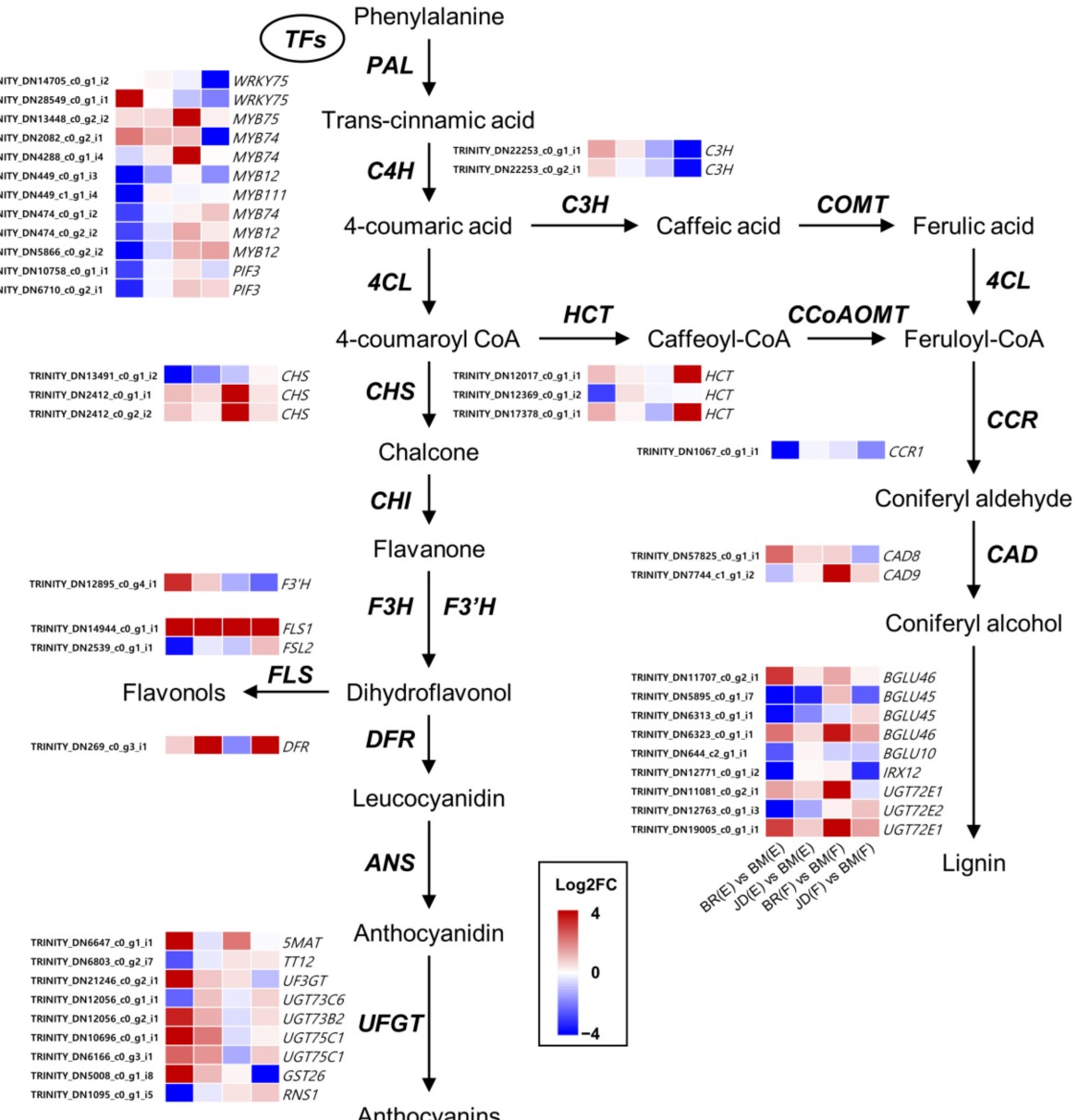

**Figure 3.** Simplified scheme and heat-maps of the differentially expressed genes (DEGs) involved in phenylpropanoid pathways. Heat-maps shown the log2Foldchange values of FPKM from RNA-seq. *PAL* Phenylalanine ammonia lyase, *C4H* Cinnamate 4-hydroxylase, *4CL* 4-coumarate:CoA ligase, *CHS* Chalcone synthase, *C3H* p-coumarate 3-hydroxylase, *CCR* Cinnamoyl CoA reductase, *F3H* Flavonone-3-hydroxylase, *F3′H* Flavonoid 3-monooxygenase, *CAD* Cinnamyl alcohol dehydrogenase, *HCT* Hydroxycinnamoyltransferase, *CCoAOMT* Caffeoyl-CoA 3-O-methyltransferase, *DFR* Dihydroflavonol 4-reductase, *ANS* Ferulate 5-hydroxylase, *UFGT* UDP-glucose: flavonoid 3-O-glycosyltransferase.

### 3.5. Validation of RNA-seq Data Using qRT-PCR

To validate the transcriptome data, selected unigenes were analyzed using qRT-PCR and the results were compared with the FPKM values from RNA-seq. The twelve structure genes of phenylpropanoid pathway including *CHS* (TRINITY_DN2412_c0_g2_i2), *F3′H* (TRINITY_DN12895_c0_g4_i1), *FSL2* (TRINITY_DN2539_c0_g1_i1), *DFR* (TRINITY_DN269_c0_g3_i1), *ANS* (TRINITY_DN14944_c0_g1_i1), *UGT73B2* (TRINITY_DN12056 _c0_g1_i1), *5MAT* (TRINITY_DN6647_c0_g1_i1), *UF3GT* (TRINITY_DN21246_c0_g2_i1), *GST26* (TRINITY_DN5008_c0_g1_i8), *UGT75C1* (TRINITY_DN6166_c0_g3_i1), *TT12* (TRIN-

ITY_DN6803_c0_g2_i7), and *RNS1* (TRINITY_DN1095_c0_g1_i5) were evaluated and the results are shown in Figure 4. First, according to the expression of genes involved in the phenylpropanoid pathway, seven unigenes were significantly up-regulated in BR mutant at the young-bud stage, including *CHS*, *F3′H*, *ANS*, *5MAT*, *UF3GT*, *GST26*, and *UGT75C1*. These results were perfectly consistent with the FPKM values from RNA-seq. For example, the correlation coefficient (r) was 0.9999 (*UF3GT*), 0.9997 (*GST26*), and 0.9994 (*5MAT*) between qRT-PCR and RNA-seq, and 9 out of 12 unigenes showed correlation values > 0.9. However, the results of qRT-PCR of the full-bloom stage in the JD mutant showed low correlation values for anthocyanins transferase genes such as *UF3GT*, *GST26*, and *UGT75C1*.

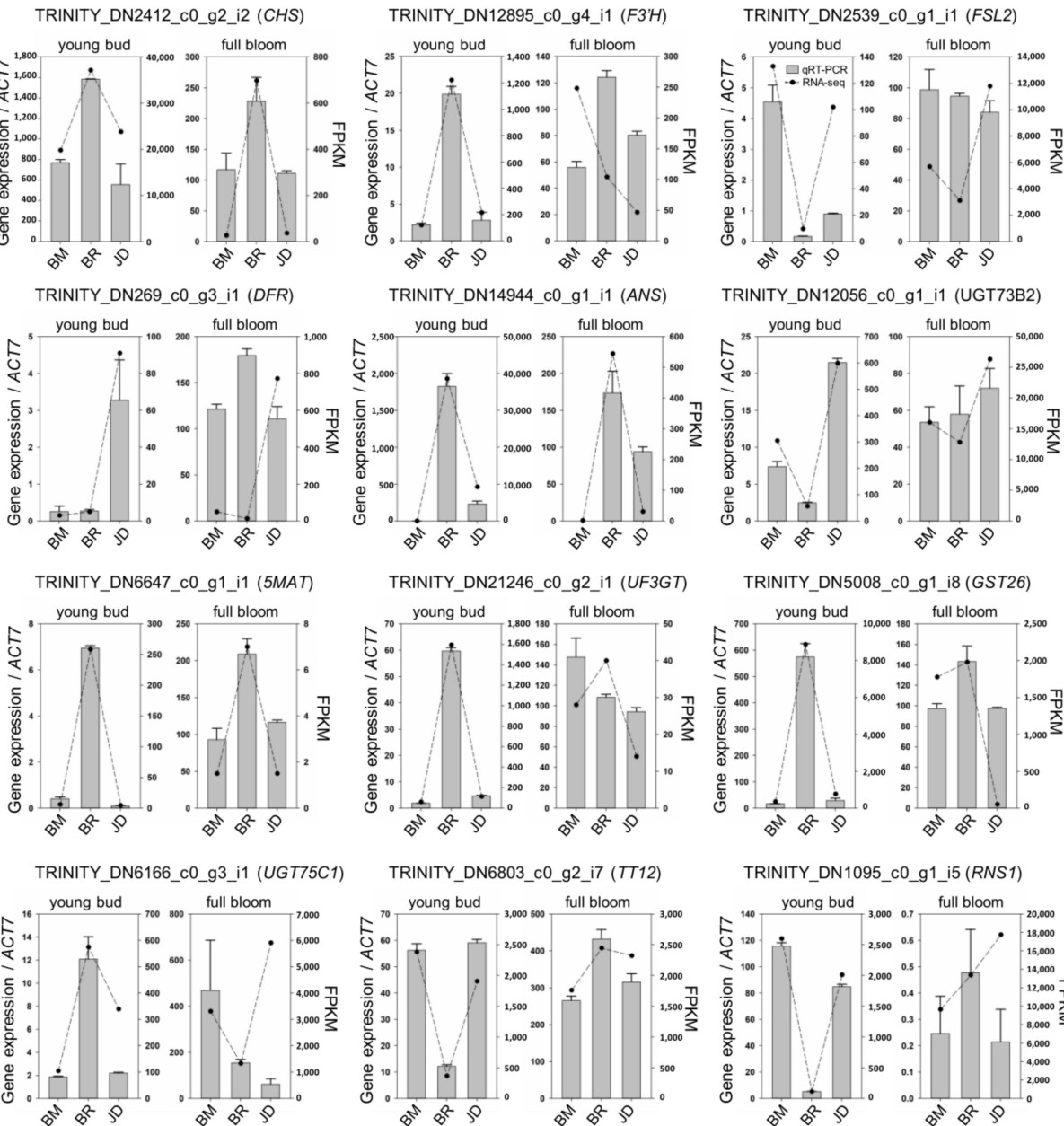

**Figure 4.** Validation of differentially expressed genes related to flavonoids and anthocyanins biosynthesis by qRT-PCR. Vertical bars are relative expression values (left *y*-axis) normalized to the reference genes (*HcACT7*) by qRT-PCR, and lines indicate expression levels calculated from FPKM values (right *y*-axis) obtained by RNA sequencing. qRT-PCR data are based on three biological replicates and represent the mean ± SE of triplicate repeats.

## 4. Discussion

Kenaf is a valuable multipurpose crop because of its high content of fiber, protein, and bioactive compounds that are beneficial to human health. Kenaf flowers, according to the cultivars, have a variety of petal colors including white, magenta, and ivory. They have been used in traditional medicine. In the present study, we determined the phenolic compound levels in the flowers of new three kenaf mutants including Baekma (white petal), Bora (purple petal), and Jangdae (ivory petal), and analyzed the transcriptome data to better understand the mechanism of flower coloration.

### 4.1. Levels of Phenolic Compounds in Kenaf Flowers

In three kenaf flower (full-bloom) variants, we investigated the six phenolic compound contents including TPC, TAC, hibiscus acid, D3S, D3G, and Q3G by using the UPLC system (Table 1). The levels of all compounds, except Q3G, were highest in BR flowers: TPC (132.98 mg/g FW), TAC (48.85 mg/g FW), hibiscus acid (12.29 mg/g FW), D3S (16.22 mg/g FW), and D3G (14.65 mg/g FW). Although Q3G was highest in JD (54.10 mg/g FW), BR flowers also showed high levels (50.90 mg/g FW) compared to BM (1.29 mg/g FW). Kenaf plant has been extensively studied for their composition of phenolic compounds because flavonoids and anthocyanins not only affect pigmentation, but also contribute to bioactive effects such as antioxidant and antibacterial activities [4,26]. Ryu et al. [27]) reported that TPC levels of the 'Auxu' kenaf cultivar flower ranged from 183.4 to 308.5 mg/100 g of dry weight using different solvents. Park et al. [28] reported the composition of phenolic compounds in flowers of 'Jangdae' and 'Jeokbong' kenaf cultivars to be 6469.9 and 6983.3 µg/g dry weight, respectively. Additionally, these previous studies also investigated the composition of TPC levels in the various kenaf tissues including leaves, stems, roots, flowers, and seeds, which showed the highest TPC levels in leaves or flowers according to the extract solvents. Hibiscus acid, as a dimethyl sulfoxide monosolvate, is widely found in the Malvaceae family and has therapeutic and pharmacological properties including $\alpha$-amylase inhibitory [29], antihypertensive [30], and antimicrobial effects [31]. However, these studies were carried out on roselle (Hibiscus sabdariffa), and studies on hibiscus acid in kenaf are limited. This study is the first to report the detecting hibiscus acid in kenaf flowers.

Anthocyanins play an important role in plant pigmentation and the TAC level was strongly associated with the coloration of the leaf and flower in kenaf. For example, previous data have shown that the TAC levels were higher in red-leaved compared to green-leaved kenaf varieties, such as BM. In BM leaves, three anthocyanins including delphinidin-3-glucoside, delphinidin-3-galactoside, and cyanidin-3-glucoside were not detected [14]. In the present study, purple flowers showed high levels of TAC and three anthocyanins, D3S, D3G, and Q3G. D3G was only detected in BR flower and is probably the key molecule involved in the formation of the purple flower. These results indicate that anthocyanins are the most important compounds for determining color of leaf/flower. For a better understanding of the genetic mechanism, transcriptome analysis was performed.

### 4.2. Transcriptome Analysis of Different Kenaf Flowers

Transcriptome analysis by RNA-seq is a useful tool for evaluating gene expression at the mRNA level and provides detailed knowledge for identifying candidate genes of biochemical traits. In kenaf, transcriptome analysis was widely used to develop molecular markers [32–34], find putative genes involved in agronomic [35,36] and biochemical traits [17,37], and screen candidate genes involved in stress response [18,38]. However, these studies were conducted using limited tissues including leaf, shoot, and stem. In the present study, using RNA-seq by paired-end sequencing technology performed analyses of the kenaf flower transcriptome. According to RNA-seq data, the three kenaf at two different developmental stages were assembled, resulting in 81,982 total transcripts with an N50 score of 1882 bp; the total was similar to the previous kenaf study [16] or achieved a slightly decreasing assembly level compared with Zhang et al. [17]. In the de novo assembly process, assembled transcript quality is more important than the number of generated

transcripts. Yang et al. [39] reported that de novo assembly of short reads causes noise in analyses and a large amount of missing data in the aligned matrix due to the redundancy, error, and incompleteness of assembled transcripts for each gene. Therefore, this study performed additional filtering with a trinity tool with an in-house script to assemble transcripts and successfully generated 38,601 representative transcripts with an N50 score of 1643 bp. Among 38,601 representative transcripts, approximately 85.64% (33,057) of the transcripts were annotated through the two public databases (Table 2).

To compare the transcriptome of three different flowers of different colors in kenaf, DEGs analysis was performed. Compared with the white flowers (BM), 1126 DEGs (958 DEG; young-bud, 168 DEGs; full-bloom) were identified in purple flowers (BR) and 390 DEGs (86 DEGs; young-bud, 304 DEGs; full-bloom) were identified in ivory flowers (JD). Among these DEGs, 756 transcripts were up-regulated and 760 transcripts were down-regulated (Figure S1). In addition, 1358 (89.6%) out of 1516 total DEGs were annotated in the NR database. Based on the results of DEG analysis, collected DEGs were displayed on a heatmap along with hierarchical clustering for a better understanding of the expression patterns in the three kenaf mutants at two different developmental flower stages (Figure 2). Furthermore, as a result of the KEGG enrichment analysis, most DEGs were strongly enriched in 'phenylpropanoid biosynthesis' within the 'biosynthesis of secondary metabolites' category.

### 4.3. Putative Candidate Genes Involved in the Phenylpropanoid Pathway

Phenylpropanoid biosynthesis is a major pathway involved in flower, leaf, and stem color formation in plants. In kenaf, the key genes that encode the phenylpropanoid pathway, including *PAL*, *CHI*, *C4H*, *F3H*, *4CL*, *FLS*, and *CHS*, have been identified in previous studies [11,13]. Additionally, 29 DEGs were previously found to be related to the biosynthesis of anthocyanins and kaempferitrin using transcriptome analysis in kenaf leaves, and these DEGs were assigned to eight structure genes including *4CL*, *CHS*, *CHI*, *F3H*, *FLS*, *DFR*, *ANS*, and *3GT* [14]. In the present study, we similarly identified 33 DEGs involved in the phenylpropanoid biosynthetic pathway, but these DEGs were assigned to 15 different types of structure genes including the 9 flavonoid and anthocyanins biosynthetic pathway genes: *CHS*, *F3′H*, *FLS*, *DFR*, *MAT*, *UFGTs*, *TT12*, *GST*, and *RNS*; and the six lignin biosynthetic pathway genes: *C3H*, *HCT*, *CCR*, *CAD*, *BGLUs*, and *IRX* (Figure 3). Furthermore, *UFGTs* and *MAT* were consistently associated with anthocyanin pigments in plants [40]. Further, our RNA-seq and qRT-PCR data showed that these genes are strongly differentially expressed in purple flowers (BR) at the young-bud stage (Figure 4). These genes are probably responsible for increased anthocyanin levels and contribute to color formation.

### 4.4. Key Regulators of Differentially Expressed Genes during Flower Coloration

Phenylpropanoid biosynthesis is regulated by a complex transcriptional network of structural genes and TFs including *WRKY*, *bZIP*, *bHLH*, *WD40*, and *R2R3-MYBs* [6]. Importantly, the R2R3-MYBs are key regulators of secondary metabolism [41] and pigmentation [42] in plants, which typically act as activators or repressors according to their C-terminal domains. However, these functions are highly variable depending on plant-specific processes [43]. MYB11, MYB12, and MYB111, members of subgroup 7 in the R2R3-MYB gene family, are transcription factors that regulate several flavonoid biosynthetic genes, including *CHS*, *CHI*, *F3H*, and *FLS* in Arabidopsis [8]. In the present study, transcripts related to *MYB12* (TRINITY_DN449_c0_g1_i3, TRINITY_DN474_c0_g2_i2, TRINITY_DN5866_c0_g2_i2), and *MYB111* (TRINITY_DN449_c1_g1_i4) were consistently down-regulated in the BR mutant in the young bud stage, whereas *MYB75* (TRINITY_DN13448_c0_g2_i2) and *MYB74* (TRINITY_DN4288_c0_g1_i4) were strongly up-regulated in the BR mutant in the full-bloom stage. These regulators probably up-regulate flavonoid biosynthesis genes including *CHS*, *F3′H*, and *FLS1*, and anthocyanin biosynthesis genes including *5MAT*, *UF3GT*, and *UGTs* in kenaf flowers.

## 5. Conclusions

In this study, comparative transcriptome and biochemical analysis were performed to explore the putative key regulatory genes in the phenylpropanoid pathway linked to flower coloration in kenaf. A total of 45 DEGs were mapped and assigned to 24 structure genes and 11 TFs in the phenylpropanoid pathway. Specifically, R2R3-MYBs were strongly regulated in several flavonoid biosynthesis genes, including *CHS*, *F3′H*, and *FLS1*, and anthocyanin biosynthesis genes, including *5MAT*, *UF3GT*, and *UGTs*. These findings will provide useful information for a better understanding of the flower color variation in kenaf.

**Supplementary Materials:** The following supporting information can be downloaded at: https://www.mdpi.com/article/10.3390/agronomy13030715/s1, Figure S1: Comparison of the DEGs in the BR and JD compared with BM; Table S1: List of primers used for validation of DEGs results by qRT-PCR; Table S2: Quality of RNA sequencing data; Table S3: Information for GO term enrichment analysis of detected DEGs.

**Author Contributions:** Conceptualization, J.I.L. and J.R.; methodology, J.R. and J.I.L.; formal analysis, J.R. and J.I.L.; investigation, D.-G.K. and J.M.K.; resources, S.-Y.K.; data curation, S.-J.K. and J.-W.A.; writing—original draft preparation, J.I.L. and J.R.; writing—review and editing, J.I.L. and S.-Y.K.; visualization, J.I.L.; supervision, S.-Y.K. and S.H.K.; funding acquisition, S.-Y.K. and J.-W.A. All authors have read and agreed to the published version of the manuscript.

**Funding:** This work was supported by the National Research Foundation of Korea (NRF) grant funded by the Korea government (RS-2022-00156231) and the research grant of the Kongju National University in 2022.

**Informed Consent Statement:** Not applicable.

**Conflicts of Interest:** The authors declare no conflict of interest.

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
