# Peer review of "Comparative Transcriptome Analysis Identified Potential Genes and Transcription Factors for Flower Coloration in Kenaf (Hibiscus cannabinus L.)"

_agronomy, doi:10.3390/agronomy13030715_

Round 1
Reviewer 1 Report
Kenaf (Hibiscus cannabinus L.) is an important medicinal crop as it contains high amounts of phytochemical compounds including polyphenols, flavonoids and anthocyanins. In this MS, Lyu et al revealed the key genes involved in phenylpropanoid pathway in flowers of three kenaf mutants by Illumina sequencing, which would be very useful for studies on regulatory mechanisms of phenylpropanoid and flavonoid biosynthesis in kenaf. The MS presents good background literature research and is well written. However, the experimental design needs to be more reasonable, and the experimental methods need to be more specific and clearer to enable other scientists to follow the results. The conclusions should be derived from reasonable experimental design and analysis. I suggest some major revisions to improve the overall quality of the MS before publication.
In general, the selected materials should reduce genetic background differences as much as possible while having obvious phenotypic differences in the comparative transcriptomics study. For example, in this study, we selected BM, BR, and JD mutant variety as research materials which is originated form C-14, Hongma300, and Jinju, respectively. If the three mutant varieties differ significantly in flower color from C-14, Hongma300 and Jinju, it seems that the results are more reliable in the comparison between BM and C-14, BR and Hongma300, and JD and Jinju than that among BM, BR and JD. In addition, why two different developmental stages (young bud and full bloom) sampled were selected for DGE analysis, and what were the differences in phenotype and compound contents between them, and the specific sampling time and method?
Figure 1b. How can there be 4 data between young-bud and full-bloom stages of BM?
Figure 1c. Why is the comparison between BR and JD missing?
Only two biological replicates in this study are not sufficient, and a summary of RNA-seq data from 12 RNA libraries should present detailed data as a separate table.
Author Response
Reviewer #1
Kenaf (Hibiscus cannabinus L.) is an important medicinal crop as it contains high amounts of phytochemical compounds including polyphenols, flavonoids and anthocyanins. In this MS, Lyu et al revealed the key genes involved in phenylpropanoid pathway in flowers of three kenaf mutants by Illumina sequencing, which would be very useful for studies on regulatory mechanisms of phenylpropanoid and flavonoid biosynthesis in kenaf. The MS presents good background literature research and is well written. However, the experimental design needs to be more reasonable, and the experimental methods need to be more specific and clearer to enable other scientists to follow the results. The conclusions should be derived from reasonable experimental design and analysis. I suggest some major revisions to improve the overall quality of the MS before publication.
In general, the selected materials should reduce genetic background differences as much as possible while having obvious phenotypic differences in the comparative transcriptomics study. For example, in this study, we selected BM, BR, and JD mutant variety as research materials which is originated form C-14, Hongma300, and Jinju, respectively. If the three mutant varieties differ significantly in flower color from C-14, Hongma300 and Jinju, it seems that the results are more reliable in the comparison between BM and C-14, BR and Hongma300, and JD and Jinju than that among BM, BR and JD.
[Response] We thank the reviewer for this valuable comment and agree with the reviewer that mutant variety would be more reliable to compare with their original cultivars. However, Jangdae and Jinju have same flower color (ivory), and Hongma300 can be growing in Korea climate, but it does not bloom1).
1) An Improved Kenaf Cultivar ‘Jangdae’ with Seed Harvesting in Korea. (2016) Korean J. Breed. Sci. 48(3):349-354
This study was originally intended to flower color variants as a follow-up study to the previous research of leaf color variants in kenaf2). However, these leaf color variants in previous study, are showed white (Baekma) and ivory (C-14, Jeokbong) flower color. For more dramatically changing the flower colors, we added BR mutant (purple petal) in present study. We also concern about different genetic background among the three mutants, so we used to control plant (BM: pigment-deficient(white) flower) in DEG analysis. Accordingly, BR vs BM and JD vs BM DEG sets are existed, while BR vs JD DEG set is not existed.
2) Transcriptome Analysis and Identification of Genes Related to Biosynthesis of Anthocyanins and Kaempferitrin in Kenaf (Hibiscus cannabinus L.). Journal of Plant Biology (2020) 63:51–62
In addition, why two different developmental stages (young bud and full bloom) sampled were selected for DGE analysis, and what were the differences in phenotype and compound contents between them, and the specific sampling time and method?
[Response] Generally, the content of secondary metabolites that determine flower coloration such as flavonoid and anthocyanins were highest level at the full bloom stage, but biosynthesis-related genes are exhibited maximum expression level in earlier stage. According to previous study, kenaf flower also reported that the expression levels of the flavonoid biosynthesis-related genes were higher at the young-bud stage3). We confirmed through preliminary qRT-PCR test that the expression levels of genes related to flavonoid synthesis were the highest in 2~3cm young-bud sizes (see below).

3) Accumulation of Kaempferitrin and Expression of Phenyl-Propanoid Biosynthetic Genes in Kenaf (Hibiscus cannabinus). Molecules (2014) 19(10): 16987–16997.
Figure 1b. How can there be 4 data between young-bud and full-bloom stages of BM?
[Response] This figure shows the number of DEGs based on the different two stages. For example, young-bud result is combined the DEGs detected in BR vs BM and JD vs BM sets. In this regard, we already described in maintext as followes: “More detailed, as a young-bud stage, 958 DEGs were uncovered in BR vs. BM, while only 86 DEGs were identified in JD vs. BM. On the other hand, as a full-bloom stage, JD vs. BM (304 DEGs) slightly more identified DEGs than BR vs. BM (168 DEGs).” (Line 167-169)
Figure 1c. Why is the comparison between BR and JD missing?
[Response] This study was conducted to identify the DEGs of purple/ or ivory flowers compared with pigment-deficient(white) flower. The reason was described in the response to comment #1.
Only two biological replicates in this study are not sufficient, and a summary of RNA-seq data from 12 RNA libraries should present detailed data as a separate table.
[Response] As suggested by the reviewer, we have added more detailed 12 RNA libraries data in Table S1(revised table number).

Reviewer 2 Report
The authors of the manuscript entitled “Comparative transcriptome analysis identified potential genes and transcription factors for flower coloration in kenaf (Hibiscus cannabinus L.)” have performed a satisfactory job.
I request the authors to provide the following changes and clarify some issues:
1. What is the rationale for selecting these mutants of the kenaf flowers? The authors should mention it in the introduction section.
2. The authors have evaluated the RNA integrity and quantity only with Nanodrop. Qubit evaluation is often superior due to specificity and sensitivity. What is the explanation for not going for Qubit evaluation?
3. Table 1: What are tR and MS? Please mention the full form.
4. Put the significance notes in table 1 while comparing TPC, TAC… by using a,b, etc.
5. Figure 1: What is the figure legend for the y-axis in figure 1b?
6. Figure 2a: Is it possible to mention the genes’ names along with the heat map?
7. Figure 2b: The text is too small, and not readable.
Author Response
Reviewer #2
The authors of the manuscript entitled “Comparative transcriptome analysis identified potential genes and transcription factors for flower coloration in kenaf (Hibiscus cannabinus L.)” have performed a satisfactory job.
I request the authors to provide the following changes and clarify some issues:
- What is the rationale for selecting these mutants of the kenaf flowers? The authors should mention it in the introduction section.
[Response] This study was originally intended to flower color variants as a follow-up study to the previous research of leaf color variants in kenaf. However, these leaf color variants in previous study (Baekma, C-14, Jeokbong), are showed white (Baekma) and ivory (C-14, Jeokbong) flower color. For more dramatically changing the flower colors, we added BR mutant (purple petal) in present study. As suggested by the reviewer, we have added mention in the introduction section as follows: “Previously, we successfully investigated the transcriptome analysis of leaf colora-tion in kenaf, which exhibited dramatic changes in eight flavonoid structural genes. The present study aimed to examine the content of phenolic compounds and perform transcriptome analysis on three different kenaf flowers including the Baekma (BM), Jangdae (JD), and Bora (BR).” (Line 69-73)
- The authors have evaluated the RNA integrity and quantity only with Nanodrop. Qubit evaluation is often superior due to specificity and sensitivity. What is the explanation for not going for Qubit evaluation?
[Response] We agree with the reviewer that nanodrop is not fully enough to check RNA integrity and quantity. We didn’t used the Qubit recommended by the reviewer, but additionally performed another integrity check process (used 2100 Bioanalyzer) by sequencing company (Seeder, Korea). The following data was sent to us before the RNA-sequencing.

To clarify this, we have changed the sentence in “2.2. RNA extraction and cDNA library construction” as follows: “RNA quantity and integrity were evaluated on a Nanodrop ND-1000 (Thermo Fisher Scientific, USA) and 2100 Bioanalyzer (Agilent, USA). (Line 90-91)
- Table 1: What are tR and MS? Please mention the full form.
[Response] tR means retention time, and MS is negative ion mass. As suggested by the reviewer, we have changed.
- Put the significance notes in table 1 while comparing TPC, TAC… by using a,b, etc.
[Response] As suggested by the reviewer, we have added the significance notes.
- Figure 1: What is the figure legend for the y-axis in figure 1b?
[Response] y-axis is “Number of DEGs”. To clarify this, we have added y-axis in figure 1b.
- Figure 2a: Is it possible to mention the genes’ names along with the heat map?
[Response] It is not possible to mention the genes’ names along with the heatmap, but you can see the Table S3 (revised table number), the gene information was shown.
- Figure 2b: The text is too small, and not readable.
[Response] We already prepared high resolution (600dpi) version, this will be provided in published on-line version.

Round 2
Reviewer 1 Report
The MS has been improved significantly and I have no more questions.
Reviewer 2 Report
I would recommend adding the bioanalyzer figures as supplementary files. The RNA Integrity number (value) for each RNA sample should be provided in the results and discussion section.